# Mechanical Characterization of Hybrid Nano-Filled Glass/Epoxy Composites

**DOI:** 10.3390/polym14224852

**Published:** 2022-11-11

**Authors:** Ali A. Rajhi

**Affiliations:** Department of Mechanical Engineering, College of Engineering, King Khalid University, Abha 61421, Saudi Arabia; arajhi@kku.edu.sa

**Keywords:** nanocomposites, glass fiber-reinforced polymer (GFRP), multiwall carbon nanotube (MWCNT), nano-silica (NS), nano-iron oxide (NFe), mechanical characterizations

## Abstract

Fiber-reinforced polymer (FRP) composite materials are very versatile in use because of their high specific stiffness and high specific strength characteristics. The main limitation of this material is its brittle nature (mainly due to the low stiffness and low fracture toughness of resin) that leads to reduced properties that are matrix dominated, including impact strength, compressive strength, in-plane shear, fracture toughness, and interlaminar strength. One method of overcoming these limitations is using nanoparticles as fillers in an FRP composite. Thereby, this present paper is focused on studying the effect of nanofillers added to glass/epoxy composite materials on mechanical behavior. Multiwall carbon nanotubes (MWCNTs), nano-silica (NS), and nano-iron oxide (NFe) are the nanofillers selected, as they can react with the resin system in the present-case epoxy to contribute a significant improvement to the polymer cross-linking web. Glass/epoxy composites are made with four layers of unidirectional E-glass fiber modified by nanoparticles with four different weight percentages (0.1%, 0.2%, 0.5%, and 1.0%). For reference, a sample without nanoparticles was made. The mechanical characterizations of these samples were completed under tensile, compressive, flexural, and impact loading. To understand the failure mechanism, an SEM analysis was also completed on the fractured surface.

## 1. Introduction

Because of their superior stiffness, strength, low density, light weight, resistance to corrosion, superior electrical properties, and ease of manufacture, fiber-reinforced polymer nanocomposites stand out from all other materials on the market and have attracted notable studies [1,2,3]. Because of their exceptional qualities, they have been widely used in a variety of applications, including the construction of buildings and sports equipment as well as in the automotive, aviation, and defense industries [4,5]. Each application, however, is subject to a unique set of circumstances, including cyclic loading, impact conditions, stretching conditions, and deformation characteristics. Investigating the behavior of these kinds of composite materials under various conditions is crucial.

Chang [6] studied the effect of carbon-fiber-reinforced composites and GFRP with the addition of MWCNTs. It was reported that the tensile strength improved by 34.7% with the addition of the MWCNTs, and the flexural strength improved by 22.16%. Markand et al. [7] investigated the effects of adding carbon nanotube (CNT) to GFRP composite laminates when subjected to interlaminar shear and flexural loading using different percentages of hardened resin (phr) (0.25, 0.5, 0.75, and 1 phr). They discovered that CNT at 0.75 phr has the best mechanical properties, improving the flexural strength and interlaminar shear strength (ILSS) by 15.7% and 9.2%, respectively.

In an experimental investigation, M.R. Ayatollahi et al. [8] investigated the effects of the MWCNT aspect ratio on the electrical and mechanical properties of epoxy/MWCNT composite plates. They established that the aspect ratio has a significant impact on the electrical and mechanical capabilities of nanocomposite materials, with smaller MWCNTs exhibiting significantly superior qualities. Laminated fiberglass/epoxy composites were examined for their mechanical, vibrational, and damping properties by M Rafiee et al. [9,10], using a variety of carbon nanofillers, such as multiwall carbon nanotubes, graphene oxide, reduced graphene oxide, and graphene nanoplatelets. According to the experimental findings, as the nanoloading increased, the damped natural frequencies and tensile characteristics of the nanocomposites also increased.

Mostovoy et al. [11] studied basalt-fiber-reinforced epoxy composites modified with nano-graphene oxide. The nano-graphene oxide was functionalized with aminoacetic acid and APTES, which enabled the functionalization. The samples were prepared with different weight percentages of the nano-graphene oxide, and various physio-mechanical tests were conducted. The results show that the 0.5 wt% graphene oxide samples have a better tensile strength (1830 MPa) when compared to the neat composite (160 MPa). The tensile modulus improved by 31% and 19% for the modified composite, whereas the flexural strength improved by only 9% and 13%.

Josh et al. [12] used carbon fiber as a reinforcement and epoxy as a matrix with nano-graphite oxide as the filler. Vacuum-assisted resin infusion molding was used to manufacture the composites. To add the nanofiller to the composite materials, two methods were used: (i) direct spraying onto carbon fiber with ethanol as a base, and (ii) mixing the nanoparticle in resin and using it to manufacture the composite. The first method of spraying is better, as the mixing of the nanoparticles included in the resin increases the density of the resin, which reduces the wettability of the fiber and may lead to more defects. They reported an improvement in transverse tensile strength of 8% and an improvement in interlaminar shear strength of 15% with the addition of the nanoparticles. However, not much improvement was observed along the longitudinal direction.

Bekeshev et al. [13] used the mineral filler ocher with epoxy resin to develop the composite material. Ocher with a size < 40 µm and these particles were first mechanically stirred and then sonicated to obtain a uniform distribution. Different samples were fabricated with different parts according to the mass of the ocher. The results show an outstanding improvement in tensile strength of 75%, a 20% improvement in tensile modulus, and an improvement in impact strength of 83%, whereas the flexural strength and flexural modulus improved by 30% and 58%, respectively. These products also led to an increase in the yield of carbonized structures from 54% to 58–76%, which led to the low flammability of the epoxy.

Investigations were completed by Jamali et al. [14] on how Graphene Oxide Nanoplatelet (GNOP) modification and Silica-GONP loading affected the mechanical characteristics of the basalt/epoxy composite. By using FTIR, STA, and Raman spectroscopy, the introduction of the silane organic chains on the surface of the GONPs was assessed. The specimens’ mechanical strength reached their maximum levels at 0.4 weight percent S-GONPs. These specimens’ tensile, flexural, and compressive strengths were higher than those of the basal/epoxy composite by 16%, 47%, and 51%, respectively. Additionally, the mechanical moduli improved. Further, it was discovered that silane alteration of the GONPs significantly changed the specimens’ mechanical characteristics. Microscopic examinations revealed that the specimens filled with nanofiller had an improved interfacial adhesion between the basalt and the matrix.

Tian et al. [15] used a matrix made of sol–gel silica/epoxy nanocomposites to enhance the interfacial characteristics between the polymers and the fibers. The interfacial adhesion was significantly increased by the silica nanoparticles, as demonstrated in both micro- and macro-mechanical studies. When compared to the carbon fiber/epoxy system, the IFSS and transverse fiber bundle tension (TFBT) strength of the carbon fiber/20 wt% nanosilica-epoxy system rose by roughly 38% and 59%, respectively. For the CF/10 wt% nanosilica-epoxy system, the ILSS of the unidirectional laminar also rose by up to 13%. These gains can be attributed to the nanoparticle-enhanced, toughened matrix, which improves stress transfer and resists debonding by reducing stress concentration and dissipating more deformation energy.

Shu-quan et al. [16] studied the tensile behavior of CNT-modified epoxy composites which included tensile modulus and tensile strength. A proportional improvement in tensile properties (strength and modulus) was reported, with an increase in filler quantity up to 1.75% of the mass fraction. The optimum mass fraction obtained was 0.75% epoxy resin. If the mass fraction increases by more than 1.75%, then the tensile strength and tensile modulus are lower than the neat resin. Similarly, Zhou et al. [17] reported an improvement in the resistance to crack propagation upon induction of the CNT in the epoxy resin nanocomposite. However, Wong et al. [18] reported that the increase in the weight percentage of multiwall carbon nanotubes (MWCNTs) in polystyrene resin has adverse effects on tensile strength, tensile modulus, and failure strain. Thus, an optimum weight percentage of nanoparticles to be added into the resin system needs to be determined, which would enhance the mechanical properties of the composite materials. Improvements in the mechanical, rheological, thermal, and adhesion properties of the nano-modified polyester with nano-silica were observed because of the formation of a hydrogen bond between the silanol groups and the ester carbonyl group on the nano-silica surface in soft segments [19,20,21,22]. According to Sudirman et al., adding nano-silica to polyester resin improves the chain mobility of the polymer, which results in an improved order compared to pure resin [23].

Zheng et al. [24] carried out a similar study in which NS was used to modify epoxy resin, and this modified resin was used as a matrix system in composites with glass fiber as reinforcement. Three different weight percentages of NS—1%, 5%, and 7%—were used for the preparation of the different laminates. Under tensile loading, the tensile strength and the tensile modulus improved by 24% and 22%, respectively, and a relatively smaller improvement was reported for the compressive and shear strengths, which improved by 13% and 14%, respectively. Under bending, the strength also increased by 22%. The reason for this improvement was given as a strong covalent bond between nano-silica and the fiber surface, which led to a better transfer of load and stress from the fiber to the matrix, and vice versa. As the weight percentage of nano-silica was high, the uniform dispersion of NS in the resin was not possible, thereby the compressive and shear properties did not improve to that extent.

Thus, the addition of nanoparticles in composite materials has both encouraging and destructive effects on the latter’s properties, which include both physical and mechanical properties. This change in the properties of a nanoscale hybrid composite depends on (i) the geometry of the nanofiller, (ii) the type of nanofiller, (iii) the type of resin system, (iv) the filler percentage, (v) the dispersion of nanoparticles in the resin system, and (vi) the manufacturing method. The present work aims to optimize the types of nanofillers and their quantities to improve mechanical properties. Three different nanoparticles were used—nano-iron oxide, nano-silica, and MWCNTs—with four different weight percentages of 0.1, 0.2, 0.5, and 1.0 wt%. Four different mechanical tests—tensile, compressive, bending, and impact—were performed. Epoxy resin was used as a matrix material and unidirectional E-glass fibers were used as reinforcement.

## 2. Material and Methods

### 2.1. Materials

As mentioned, Lapox L12 is an epoxy resin that is commercially available along with a K-6 hardener and was used in the preparation of the laminates due to its wide utilization in the industry. The chemical name of Lapox L12 is Diglycidyl Ether of Bisphenol. Table 1 displays the various properties of the Lapox L12 used in this work. Three different nanoparticles, as mentioned earlier, were supplied by Intelligent Materials Pvt. Ltd., Punjab, India. and were used as nanofillers. Later, these particles were subjected to SEM (Zeiss GeminiSEM 360, ZEISS Microscopy, Jena, Germany) analysis for quality assurance and characterization, which are discussed in detail in Section 3. Figure 1 shows the SEM images of these nanoparticles, and with this analysis, the size and the other specifications presented in Table 1 were determined. A 600 GSM E-glass fiber roll containing 34% chopped strand fiber and 66% unidirectional fiber was used as reinforcement.

Figure 1 represents the schematic diagram of the fabrication of the modified fiber-reinforced polymer nanocomposites. Nanofillers were dispersed in a resin system in two stages, the first including mechanical stirring for an hour followed by bath sonication for an hour. Later, this modified resin system was used in the preparation of the samples using a hand lay-up technique followed by a compression molding technique at a pressure of 20 MPa. The samples required for the tensile, compression, bending, and impact tests were cut according to the ASTM (D3039, 695, D790 & D265) standard from a single laminate with a size of 250 × 250 mm^2^. The same procedure was opted for the preparation of the different laminates from the different nanoparticles according to size and concentration, type of nanoparticle, and weight concentration, as tabulated in Table 2.

### 2.2. Mechanical Characterizations

The mechanical properties of the samples were determined under tensile, compression, bending, and impact loading. The tensile tests were performed per the ASTM D3039 standard with a crosshead speed of 1 mm/min on the universal testing machine (UTM), Deepak Poly Plast, with a range of 5 tons. The specimens were cut from a single laminate with a width of 20 mm, a thickness of 2.5 mm, and a length of 240 mm. The gauge length of the samples was 100 mm with a clamping length of 50 mm. Three specimens from each sample were tested.

A static compression test was conducted on all samples using the UTM with a modified end-loading fixture as proposed by Shimokawa per the ASTM standard 695. The specimen was tightly calmed in the fixture with binding strips. The fixture has a pair of supporting guides that prevent the out-of-plane directional movement of the specimen under loading. The binding strips were used to apply the compression load at the end of the sections of the specimens.

Flexural tests under 3-point bending were also completed using a UTM per ASTM D790 to calculate the flexural properties (strength and modulus). The crosshead of the UTM moved with a constant speed of 2.0 mm per min and with a depth-to-span ratio of 1:16. The slope of the load-displacement graph was used to calculate the flexural modulus. The flexural strength of the specimen is the highest stress at failure on the tensile side. An average of three specimens were taken.

The Izod impact testing method was used to determine the energy observed (impact strength) for the breaking of the fabricated composite specimens per ASTM D256. Digital impact testing was used, which has advantages like more versatility, ease of operation, and the display of information with a high resolution. The machine has a maximum pendulum capacity of 25 J, a drop height of 0.61 m, and an impact velocity of 3.46 m/s. The specimen used was 8 cm × 3 cm × 0.4 cm. The specimens were cut along the longitudinal directions only.

## 3. Results and Discussion

### 3.1. SEM Analysis of the Nanoparticles

An SEM analysis of the nanoparticles was completed using a ZEISS GeminiSEM 360 machine. The images are shown in Figure 2 and the details of the results are tabulated in Table 3.

### 3.2. Tensile Results

Figure 3 represents the load-displacement graphs under tensile loading of the neat glass-fiber-reinforced polymer (GFRP) and fiber-reinforced polymer nanocomposite GFRP with the different nanoparticles (NFe, MWCNT, NS) under different weight concentrations. From Figure 3a, it is observed that the breaking point of the fiber-reinforced nanocomposite with NFe as the filler under tensile load was earlier than that of the neat GFRP, except for the 0.1 wt% NFe sample, and this is because of the increase in the brittleness of the composite materials. The area under the curve of load-displacement is an indication of the modulus of toughness, which was calculated for all samples, and it was observed that the sample with 0.1 wt% NFe was 84.6% higher when compared to the neat GFRP. Whereas, for the other sample, this value is reduced to 29.4% when compared with the neat GFRP. It can also be observed that, as the weight percentage increased, the breaking point decreased. However, for the 0.1 wt% nano-iron oxide particles, there was an extended breaking point, and there was an increase in toughness of 84.6%, which was calculated as the area under the curve. There was a reduction in toughness of 29.4% for the 0.5 wt% nano-iron oxide fiber-reinforced polymer nanocomposite sample. Regarding the breaking point of the samples, in Figure 3b, it is observed that it is the same for all samples except for the 0.1 wt% MWCNT sample. The toughness improved by 14.6%, 30.82%, 52.66%, and 78.06% for the 0.1, 0.2, 0.5, and 1.0 wt% samples, respectively. Figure 3c shows that the breaking point extended by 20% for the 0.1 wt% nano-silica fiber-reinforced polymer nanocomposite sample, and thereafter it decreased for the rest of the samples. Breaking occurred 40% earlier when compared to the neat GFRP composite material. There were improvements in toughness of 43%, 83%, 63%, and 38.1% for the 0.1, 0.2, 0.5, and 1.0 wt% samples, respectively, of the fiber-reinforced polymer nanocomposites with nano-silica as the filler. From these graphs, the tensile strengths of the specimens were obtained, and they are compiled in Figure 4a–c. The common behavior of an increase in ultimate tensile strength for the 0.1 wt% sample was observed in all the modified GFRPs. In the case of the sample modified with MWCNTs, a slight decrement in ultimate tensile strength was observed for the 0.2 wt% sample, and thereafter it remained constant. For the sample of 0.1 wt%, a tensile strength improvement of 30% overall was seen. In the case of the samples modified with nano-iron oxide particles, the maximum tensile strength of the 0.2 wt% sample exhibited an increment of 20% when compared with the neat GFRP. For the third set of samples modified with nano-silica, the maximum tensile strength for the 0.5 wt% sample exhibited an increment of 35% when compared with the normal GFRP. The good dispersion and exfoliation of the nanoparticles are the reasons for this improvement in tensile strength. The presence of an agglomerate caused a drop in the tensile strength of the specimens with a higher wt% of nanoparticles. This causes stress concentration rather than load transfer, and failure starts at this point.

### 3.3. Compression Results

As described in the above section, the compression test was carried out per ASTM D695, with a sample size of 25 × 25 mm^2^. The ultimate compressive strengths for the different weight percentages of the nanoparticle-fiber-reinforced nanocomposites are plotted in Figure 5a–c. The inclusion of the MWCNTs increased the ultimate compressive strength (UCS) for the MWCNT-set fiber-reinforced polymer nanocomposites, as seen in Figure 5a. For the 0.1 weight percent MWCNT sample, the UCS was 1.27 times higher than the clean conventional composite, and for the 0.2 weight percent MWCNT sample, the UCS increased by 1.36 times. For the 0.5 wt% and 1.0 wt% MWCNT samples, the percentage increase was almost the same, which was 26% when compared with the neat conventional composite. A proportionate increase in the UCS for the first two sets of samples was due to improvements in the resin properties with the presence of MWCNTs. As in compression, the load was mostly taken up by the matrix material, and with the addition of MWCNTs, the resin properties increased, thereby the UCS also improved up to the 0.2 wt% sample. However, with further increases in the weight percentage of MWCNTs in the fiber-reinforced polymer of 0.5 wt% and 1.0 wt%, a reduction in the UCS was observed, as a higher wt% of MWCNTs led to a decrease in the dispersion of the nanoparticles due to an increase in the van der Waals force between them.

For the samples with NFe, in Figure 5b, the maximum increase in UCS was observed for the 0.5 wt% sample, which was 37% higher than that of the conventional neat composite, and the minimum was observed in the 1.0 wt% sample, which was still 22% higher than the conventional neat composite. The volume of the nanoparticles is what caused the difference in behavior between the MWCNT sample and the nano-iron oxide samples. Low-volume nano-iron oxide is needed for the same weight of MWCNTs because nano-iron oxide has a higher density. Therefore, compared to the MWCNT samples, a shift from 0.2 wt% to 0.5 wt% was observed for the maximal strength in the nano-iron oxide samples. For the nano-silica set of samples, a parabolic improvement in UCS was observed from the neat conventional composite to the 1.0 wt% sample. The maximum UCS was observed in the maximum wt% sample—that is, the 1.0% sample—and was 74% higher than the conventional composite, as seen in Figure 5c. No decreasing tendency was seen in the other two sets of samples, mainly because of the same molecular formula of glass fiber and nano-silica. This resulted in the addition of supporting reinforcement and glass fibers. The agglomeration that occurred at a high weight percentage was used to distribute the load rather than acting as a stress concentration point.

### 3.4. Flexural Results

The typical load-deflection curves for the neat GFRP and fiber-reinforced polymer nanocomposite GFRP under 3-point bending are shown in Figure 6. From these graphs, a clear indication of improvement in the flexural modulus of all the fiber-reinforced polymer nanocomposite GFRPs when compared to the neat GFRP composite can be observed from the slope of these curves. The flexural modulus of all the specimens was determined from these graphs using the equation
$$E\;=\frac{{\mathrm{ml}}^{3}}{{\mathrm{bt}}^{2}}$$
where *E* is the flexural modulus, *m* is the slope of the curve, *b* is the width of the specimen, *l* is the gauge length, and *t* is the thickness of the specimen. The plastic region for all sets of the fiber-reinforced polymer nanocomposites with MWCNTs is very small, that is, the breaking occurred immediately after the elastic limit as shown in Figure 6a. For the samples of the neat GFRP, there was a prolonged plastic stage before breaking, which is witnessed from the nonlinear regions of the graphs. The maximum improvement in flexural modulus was 87% for the sample with 0.2 wt% of MWCNTs, and the minimum was for the 1.0 wt% MWCNT sample, which was 49.8% higher when compared with the neat GFRP. The energy stored in the sample within the elastic limit is the area under the curve and is 72% higher in the 0.1 wt% MWCNT fiber-reinforced polymer nanocomposite sample when compared with the neat GFRP. In the rest of the samples, even though there was an increase in the flexural modulus, due to early breaking, there was no variation in the energy stored by the samples when compared to the neat GFRP. In the fiber-reinforced polymer nanocomposites with nano-iron oxide, an improvement in the flexural modulus of all the samples similar to the MWCNT FGCMs can be observed in Figure 6b. The flexural modulus improved by 62%, 72%, 66%, and 69% for the 0.1, 0.2, 0.5, and 1.0 wt% samples, respectively, when compared with the neat GFRP sample. Nonlinear regions do exist in these samples, which indicates a bit of plastic deformation before breaking. The energy stored in the nano-iron oxide sample within the elastic limit upon loading was the same as the neat GFRP sample, except for the 1.0 wt% sample, which was 68% higher. Figure 6c shows the load-deflection graphs for the fiber-reinforced polymer nanocomposites with nano-silica and the neat GFRP. Due to the addition of this nano-silica, improvements in flexural modulus, flexural strength, and energy storage were witnessed. The flexural modulus improved by 85% in the 0.5 wt% nano-silica fiber-reinforced polymer nanocomposite sample. The energy stored was 54% higher in the 0.2 wt% nano-silica sample when compared with the neat GFRP sample. The flexural strength of all the samples in the different sets is plotted as shown in Figure 7a–c. From these graphs, it can be said that, with the increase in the weight percentage of MWCNTs, the bending strength increased for the 0.1 wt% and 0.2 wt% samples, and for the remaining two samples, it dropped and remained constant. For the 0.1 wt% and 0.2 wt% MWCNT modified GFRP samples, a 58% and 63% increase in flexural strength, respectively, was observed. Further, for the remaining two sets of samples, the increases in flexural strength were 40% and 43%, respectively. The increase in the flexural properties with the addition of MWCNTs is due to improvements in the compressive properties of the resin. The related fiber-reinforced composite’s bending strength rose as a result of this improvement. The change in flexural strength that can be seen in Figure 7b was due to the addition of iron oxide. The percentage increase in flexural strength for these samples, when compared to the unmodified GFRP composite, was 24%, 27%, 47%, and 55%. The following reasons account for the behavior difference between the MWCNT and nano-iron oxide sets of samples: (a) Because the density of iron oxide is higher than that of MWCNTs, the dispersion in epoxy resin is much better in the former than in the latter; and (b) the spherical shape of nano-iron oxide replaces the defects caused by air bubbles, due to which they act as a stress-transfer medium in the laminate. For GFRP modified with NS, the behavior is very similar to the samples modified with nano-iron oxide, which can be observed in Figure 7c. The increase in flexural strength when compared to the conventional composite was 52%, 62%, 64%, and 94% in the order of increasing weight percentage of the samples.

### 3.5. Impact Result

As discussed in Section 3, the Izod impact test was executed, and the impact strength of the samples with diverse nanoparticles such as MWCNTs, NFe, and NS can be seen in Figure 8a–c. For the MWCNT set of samples: (i) the 0.1 wt% sample had the highest impact strength; and (ii) for the 1.0 wt% sample, the rise in impact strength was 66%, and the minimum rise was 26%. The impact strength of the nano-iron oxide sample increased by 127% for the 0.5 wt% sample, reached a maximum of 89% for the 0.1% sample, and reached a minimum of 54% for the 1.0 wt% sample in the case of the samples with nano-silica particles. The escalation in impact strength of the aforesaid lamina is a consequence of morphological modification in the resin during crystallization. The impact strength reduced and remained constant thereafter for the 0.2 wt% sample in the case of the samples with MWCNTs and nano-silica compared to the nano-iron oxide samples because of the agglomeration of these particles in the resin base.

### 3.6. SEM Analysis

An SEM analysis of the fractured surface of a specimen subjected to tensile and flexural loading was conducted. Figure 9a shows an SEM image of the modified GFRP sample with 0.5 wt% nano-silica, where a fiber pullout phenomenon is not observed and the fibers are broken. This suggests that the presence of nano-silica in the FRP caused an increase in friction for the fibers to come out. Figure 9b shows an agglomeration of nanoparticles, which led to a reduction in tensile strength in the case of the 0.5 wt% MWCNT sample. For the bending-test sample, a rough surface is observed on the fractured area as shown in Figure 9c. This indicates delamination and fiber splitting.

## 4. Conclusions

This article studies the mechanical behavior of modified GFRP with nanoparticles with different weight percentages. The maximum tensile strength was found in the 0.5 wt% nano-silica modified GFRP. For all the types of specimens, the ultimate tensile strength decreased with the increasing addition of the nanoparticles because of agglomeration. It is found that the highest percentage gain in compressive strength was 89% for the sample modified with nano-iron oxide (the 0.5 wt% sample). It can also be observed that the nanoscale composites were more brittle than the normal composites, thereby the breaking point was substantially earlier in all samples when compared to the plain resin. With the inclusion of nanoparticles, the impact strength was also enhanced; the largest increase was 127% for the sample modified with nano-iron oxide. In the case of flexural properties, the maximum strength was found in the nano-silica samples with 1.0 wt%. The research on oriented MWCNTs and GF/EP composites reinforced with film-shaped MWCNTs is still ongoing.

## Figures and Tables

**Figure 1 polymers-14-04852-f001:**
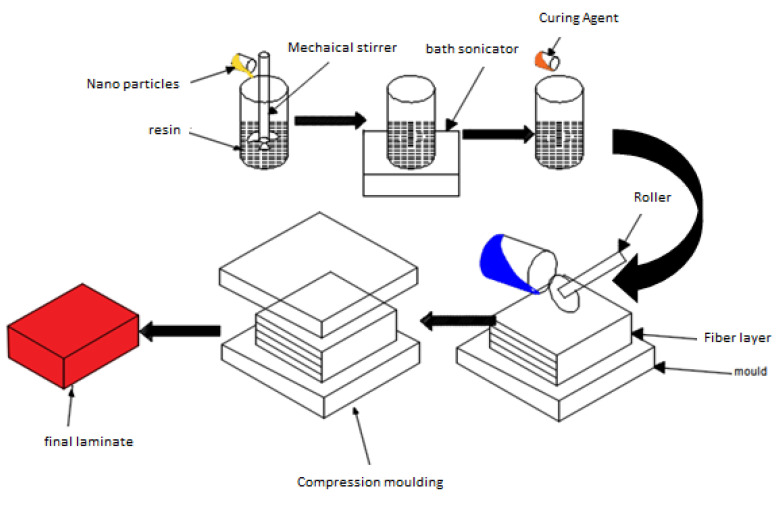
Schematic view of the fabrication of modified nano-polymer composites.

**Figure 2 polymers-14-04852-f002:**
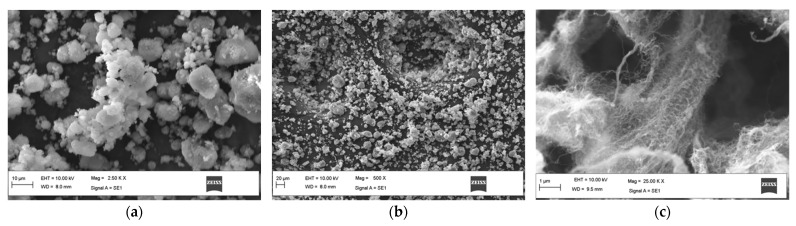
SEM images of (**a**) Nano-iron oxide, (**b**), Nano-silica, and (**c**) MWCNTs.

**Figure 3 polymers-14-04852-f003:**
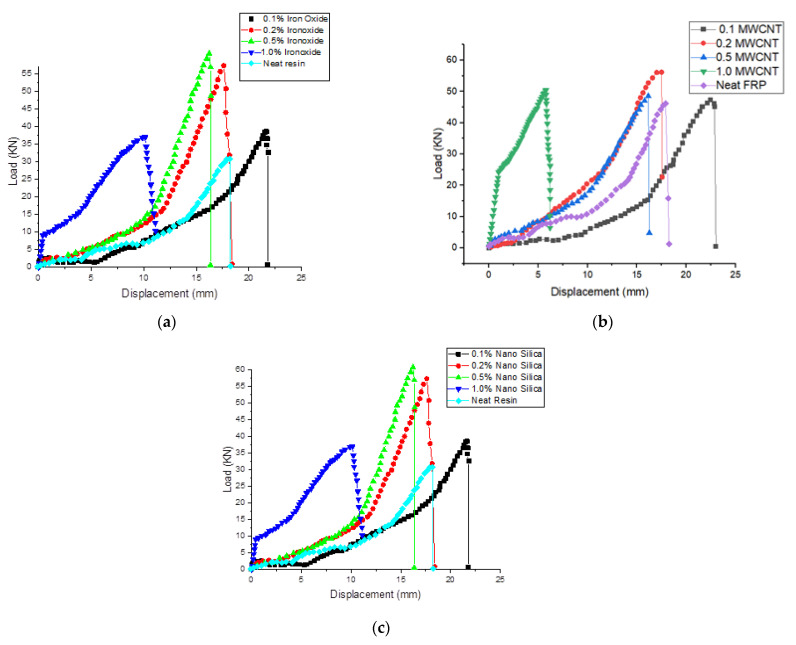
Load-displacement graphs of tensile tests for modified GFRP with (**a**) NFe, (**b**) MWCNTs, and (**c**) NS.

**Figure 4 polymers-14-04852-f004:**
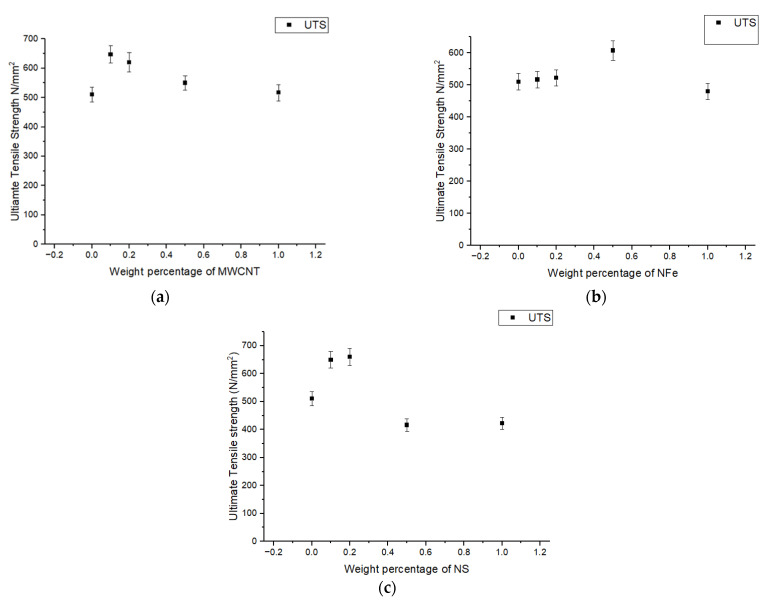
Variation in ultimate tensile strength of modified GFRP samples with different weight percentages of (**a**) MWCNTs, (**b**) NFe, and (**c**) NS.

**Figure 5 polymers-14-04852-f005:**
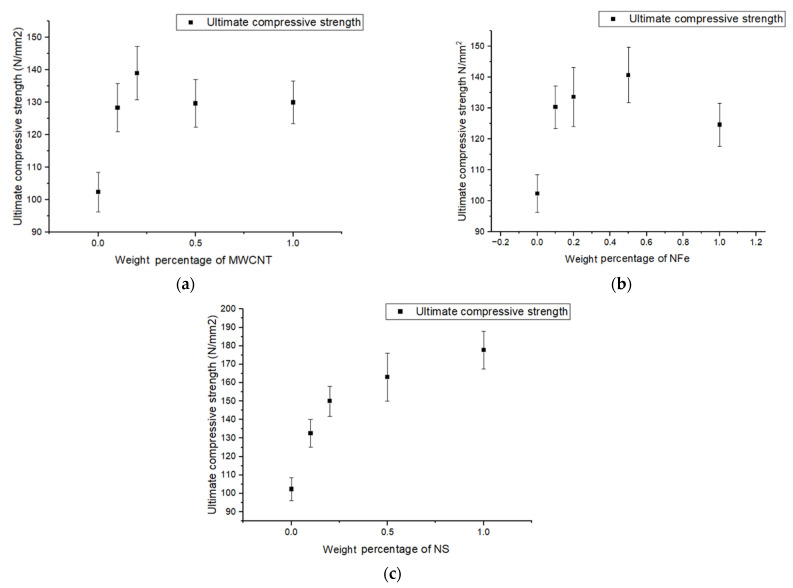
Variation in ultimate compressive strength of modified GFRP samples with different weight percentages of (**a**) MWCNTs, (**b**) NFe, and (**c**) NS.

**Figure 6 polymers-14-04852-f006:**
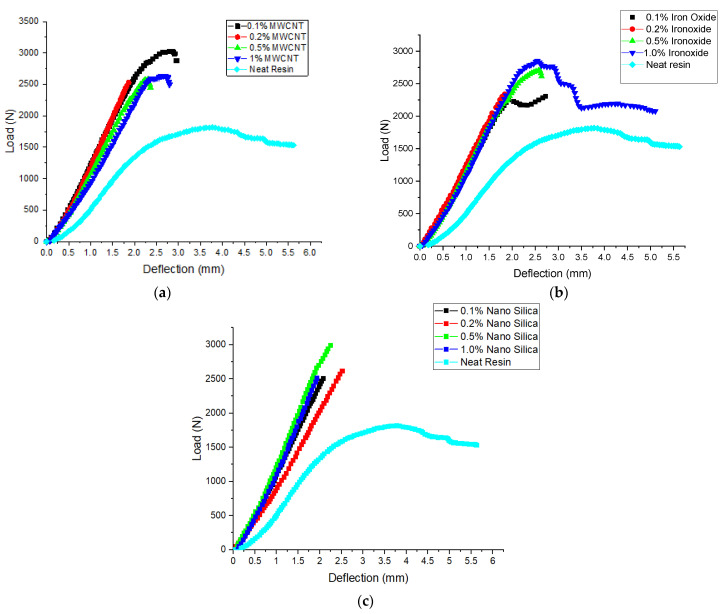
Load vs. deflection graphs of bending tests for modified GFRP with (**a**) MWCNTs, (**b**) NFe, and (**c**) NS.

**Figure 7 polymers-14-04852-f007:**
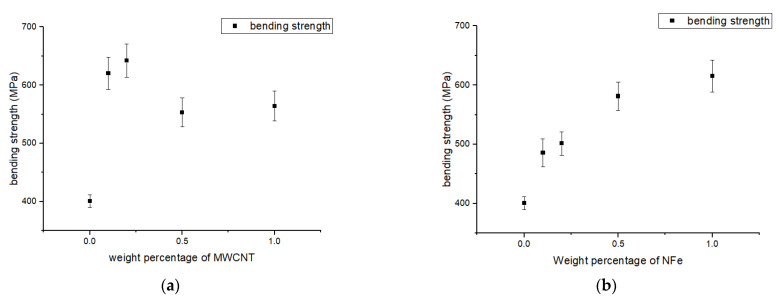
Variation in flexural (bending) strength of modified GFRP samples with different weight percentages of (**a**) MWCNTs, (**b**) NFe, and (**c**) NS.

**Figure 8 polymers-14-04852-f008:**
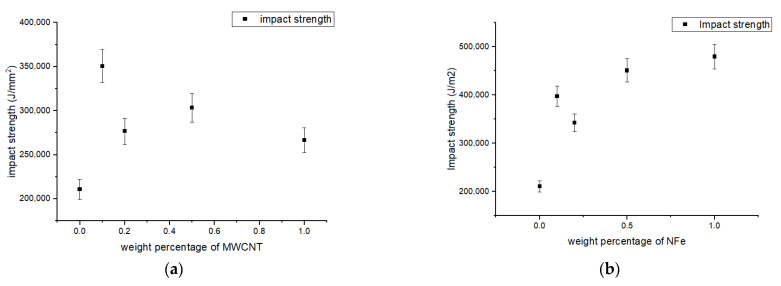
Impact energy stored under Izod impact test of modified GFRP samples with different weight percentages of (**a**) MWCNTs, (**b**) NFe, and (**c**) NS.

**Figure 9 polymers-14-04852-f009:**
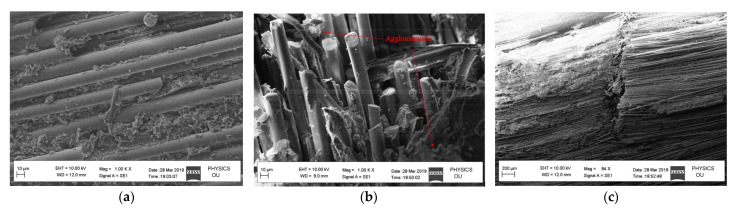
SEM photo of samples: (**a**) tensile load for 0.5 wt% NS, (**b**) tensile load for 0.5 wt% MWCNTs, and (**c**) flexural load for 0.5 wt% NFe.

**Table 1 polymers-14-04852-t001:** Properties of Lapox L12.

S. No	Properties	Value
1.	Appearance	Clear, viscous liquid
2.	Viscosity at 25 °C	9000–12,000 m Pas
3.	Specific gravity at 25 °C	1.1–1.2
4.	Solubility	30 g/25 mL
5.	Melting point	88–92 °C
6.	Pot life	6 h–8 h at 20 °C 5 h–7 h at 30 °C 3 h–5 h at 40 °C
7.	Tensile strength	70–80 MPa
8.	Elastic modulus in tension	4.0–4.8 GPa
9.	Glass transition temperature (DSC)	150–160 °C
10.	Co-efficient of linear thermal expansion	45–55 × 10^−6^ K^−1^

**Table 2 polymers-14-04852-t002:** Different compositions used in sample preparation.

S. No	Type of Resin	Type of Nanoparticle	Weight Percentage of Nanoparticles
1	EPOXY	MWCNT	0.10%
2	EPOXY	NS	0.10%
3	EPOXY	NFe	0.10%
4	EPOXY	MWCNT	0.20%
5	EPOXY	NS	0.20%
6	EPOXY	NFe	0.20%
7	EPOXY	MWCNT	0.50%
8	EPOXY	NS	0.50%
9	EPOXY	NFe	0.50%
10	EPOXY	MWCNT	1.00%
11	EPOXY	NS	1.00%
12	EPOXY	NFe	1.00%
13	EPOXY	-	-

**Table 3 polymers-14-04852-t003:** Properties of nanoparticles.

Material Properties	Nano-Silica	Nano-Iron Oxide	MWCNT
Size (nm)	50	50	30–50
Shape	Spherical	Spherical	Cylindrical
Purity	99.9%	99.9%	99.8%
Color	White	Reddish	Black

## Data Availability

Not applicable.

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
