# Peer review of "Mechanical Characterization of Hybrid Nano-Filled Glass/Epoxy Composites"

_polymers, 2022, doi:10.3390/polym14224852_

Round 1

Reviewer 1 Report

11.      The abstract must be reduced at about 200 words

22.   The introduction is very extensive. It should be more focused  

3.      The originality must be emphasized and the differences should have been highlighted and valued;

4.      To replace the title of chapter 2: Experimentation with Material and Methods according to instruction for authors.

5.      All the results included in chapter 2 must be transferred to the chapter Results and discussions in its proper place.

6.      To specify the exact name and type of the devices used in the study and their working parameters (SEM, UTM, etc.)

7.      To include chapter 3. Mechanical Characterization as component of chapter 2, thus chapter 4. Results and discussions becomes chapter 3.

8.      Some sentences should be reformulated in accordance with the grammatical rules and for an appropriate meaning, avoiding the repetition of some words in the same sentence (ex. sentences from lines 176-181, 224-225, ….)

9.      In the row 224 the “vanderwaal force” must be replaced with “van der Waals force”.

10.   To give more extensive explanations of each category of determined properties and the correlation between them.

11.  All the references must be written according to the authors guide

12.   The references should also be updated with articles published in the 2012-2022 decade

Author Response

Point 1: The abstract must be reduced at about 200 words.

Response:

The abstract has been reduced to about 200 words.

Point 2: The introduction is very extensive. It should be more focused.

Response:

 The introduction has been reworked to be more focused on the topic. Please see the modified version of the manuscript.

Point 3: The originality must be emphasized and the differences should have been highlighted and valued;

Response:

The originality and the differences are now addressed in the modified introduction.

Point 4: To replace the title of chapter 2: Experimentation with Material and Methods according to instruction for authors.

Response:

It is corrected now in the revised manuscript.

Point 5: All the results included in chapter 2 must be transferred to the chapter Results and discussions in its proper place.

Response:

It is corrected now in the revised manuscript.

Point 6: To specify the exact name and type of the devices used in the study and their working parameters (SEM, UTM, etc.)

Response:

The tensile tests were performed as per ASTM D3039 standard with a crosshead speed of 1mm/min, on the universal testing machine (UTM) Deepak poly past, range of 5 ton.

The SEM analysis of the nanoparticles was carried using Zeiss GeminiSEm 360 machine.

The Izod impact testing method is used to determine the energy observed (impact strength) for breaking of the fabricated composite specimens as per ASTM D256.

A digital impact testing is used which has advantages like more versatile, ease of operation and display of information with high resolution. The machine has a maximum pendulum capacity of 25 joules, drop height of 0.61 meters, impact velocity of 3.46 m/s. The specimen used was of the size 8cmx 3cm x 0.4 cm. The specimens are cut along the longitudinal directions only.

Point 7: To include chapter 3. Mechanical Characterization as component of chapter 2, thus chapter 4. Results and discussions becomes chapter 3.

Response:

All chapters were changed to match the suggested format.

Point 8: Some sentences should be reformulated in accordance with the grammatical rules and for an appropriate meaning, avoiding the repetition of some words in the same sentence (ex. sentences from lines 176-181, 224-225, ….)

Response:

The whole manuscript has been edited to eliminate such grammatical errors and repetition.

Point 9: In the row 224 the “vanderwaal force” must be replaced with “van der Waals force”.

Response:

Words has been edited as seen in line 395.

Point 10: To give more extensive explanations of each category of determined properties and the correlation between them. 

Response:

The explanation is addressed now in the revised manuscript.

Point 11: All the references must be written according to the authors guide.

Response:

References are now updated and written accordingly to the guide. 

Point 12: The references should also be updated with articles published in the 2012-2022 decade.

Response:

References are now updated and written accordingly to the guide.

Reviewer 2 Report

The manuscript under the title: “Mechanical Characterization of hybrid nano filled Glass/Epoxy composites” is in line with Polymers journal. This topic is relevant and will be of interest to the readers of the journal. It based on original research. This research has scientific novelty and practical significance. The article has a typical organization for research articles.
Before the publication it requires significant improvements, especially:

 1. The "Introduction" section needs to be substantially reworked. You must remove information that is not specifically relevant to the disclosure of your experimental work. The information and sources provided should substantiate the relevance of your particular experimental study directly. It is necessary to consider in more detail the issues of modifying epoxy composites with various fillers. It has been proven that the effect of fillers on the properties of polymer composites is determined by many factors: ……. I think the related references should be cited corresponding to each aspect, e.g. (but not limited to these), which will undoubtedly improve the "Introduction" section: 

  • Polymers 202214(2), 338; https://doi.org/10.3390/polym14020338
  • J. Compos. Sci. 2019, 3(1), 30; https://doi.org/10.3390/jcs3010030
  • Polymers 202012(7), 1437; https://doi.org/10.3390/polym12071437
  • Polymer Composites 2018, 39, E2472–E2482, doi:10.1002/pc.24766
  • Russ. J. Appl. Chem. 86, 765–771 (2013). https://doi.org/10.1134/S107042721305025X
  • Composites Part A: Applied Science and Manufacturing 2017, 98, 1–8, doi:10.1016/j.compositesa.2017.03.007

2. The "Introduction" section: show what is the scientific novelty of your research and how it differs from those described in the literature.

3. Section 2.1. It is necessary to add the physicochemical characteristics of components - give a table with the main physicochemical and technological properties of epoxy resin and hardener.

4. It is necessary to indicate the pressing pressure that was used in the compression molding of the composites.

5. Error bar should be added in Figure 4, 5, 7, 8.

6. Figure 3 and 4 should be moved to section 4.1.

7. Figure 6 and 7 should be moved to section 4.3.

8. Check the numbering of the figures in the figure captions and in the text of the article.

9. It would be good to see how the combination of several nanoparticles you use will affect the physico-mechanical properties of reinforced composites. In this case, a synergistic reinforcing effect is possible, since different nanoparticles provide a different mechanism for strengthening the polymer matrix.

Author Response

Point 1: 1. The "Introduction" section needs to be substantially reworked. You must remove information that is not specifically relevant to the disclosure of your experimental work. The information and sources provided should substantiate the relevance of your particular experimental study directly. It is necessary to consider in more detail the issues of modifying epoxy composites with various fillers. It has been proven that the effect of fillers on the properties of polymer composites is determined by many factors: ……. I think the related references should be cited corresponding to each aspect, e.g. (but not limited to these), which will undoubtedly improve the "Introduction" section: 

Polymers 2022, 14(2), 338; https://doi.org/10.3390/polym14020338

  1. Compos. Sci. 2019, 3(1), 30; https://doi.org/10.3390/jcs3010030

Polymers 2020, 12(7), 1437; https://doi.org/10.3390/polym12071437

Polymer Composites 2018, 39, E2472–E2482, doi:10.1002/pc.24766 

Russ. J. Appl. Chem. 86, 765–771 (2013). https://doi.org/10.1134/S107042721305025X

Composites Part A: Applied Science and Manufacturing 2017, 98, 1–8, doi:10.1016/j.compositesa.2017.03.007 

Response:

The introduction is reworked to emphasize the significance of this study and related articles have been cited in the revised manuscript to address all comments. 

Point 2: The "Introduction" section: show what is the scientific novelty of your research and how it differs from those described in the literature.

Response:

 The introduction is reworked to emphasize the significance of this study. Please refer to the revised manuscript. 

Point 3: Section 2.1. It is necessary to add the physicochemical characteristics of components - give a table with the main physicochemical and technological properties of epoxy resin and hardener.

Response:

The properties of epoxy resin L12 used in this study is now listed in Table 1 in the revised manuscript.

Table 1 properties of Lapox L12

S.No

Properties

Value

1.

Appearance

Clear, viscous liquid

2.

Viscosity at 25°C

9,000 - 12,000 m Pas

3.

Specific gravity at 25°C

1.1 - 1.2

4.

Solubility

30 g / 25 ml

5.

Melting point

88 – 92 °C

6.

Pot life

6 hours - 8 hours at 20°C

5 hours - 7 hours at 30°C

.3 hours - 5 hours at 40°C

7.

Tensile strength

70 – 80 MPa

8.

Elastic modulus in tension

4.0 - 4.8 GPa

9.

Glass transition temperature (DSC)

150 – 160 ºC

10.

Co-efficient of linear thermal expansion

45 - 55 X 10-6 K -1

Point 4: It is necessary to indicate the pressing pressure that was used in the compression molding of the composites.

Response:

Pressing pressure is now indicated in the revised version.

“Nano-fillers are dispersed in a resin system in two stages first by mechanical stirring for an hour followed by bath sonication for an hour. Later this modified resin system was used in the preparation of samples using hand lay-up technique followed by a compression moulding technique at a pressure of 20 MPa.”

Point 5: Error bar should be added in Figure 4, 5, 7, 8.

Response:

Error bars have been added in Figures 4,5,7,8

Point 6: Figure 3 and 4 should be moved to section 4.1.

Response:

Fig.3 and Fig.4 are now moved to the results and discussion section under 3.2 Tensile Results.

Point 7: Figure 6 and 7 should be moved to section 4.3.

Response:

Fig.6 and Fig.7 are now moved to the results and discussion section under 3.4 Flexural Results.

Point 8: Check the numbering of the figures in the figure captions and in the text of the article.

Response:

All figures are now numbered and cross-referenced accordingly.

Point 9: It would be good to see how the combination of several nanoparticles you use will affect the physico-mechanical properties of reinforced composites. In this case, a synergistic reinforcing effect is possible, since different nanoparticles provide a different mechanism for strengthening the polymer matrix.

Response:

Nanoparticles combination will be highly recommended for the next research work epically with the types of nanoparticles used in this study. So it will be our next research scope with highlighting their mechanical properties based on different compositions. 

Round 2

Reviewer 2 Report

The authors considered most of the comments or adequately responded to the remarks contained in the review; therefore, the work may be approved for publication.